# Resolution of Optimal Mitochondrial and Nuclear DNA Enrichment in Target-Panel Sequencing and Physiological Mitochondrial DNA Copy Number Estimation in Liver Cancer and Non-Liver Cancer Subjects

**DOI:** 10.3390/cancers16173012

**Published:** 2024-08-29

**Authors:** Xue-Ying Lyu, Yu-Man Tsui, Ivan Ka-Kit Tam, Po-Man Li, Gary Cheuk-Hang Cheung, Joyce Man-Fong Lee, Irene Oi-Lin Ng, Daniel Wai-Hung Ho

**Affiliations:** 1State Key Laboratory of Liver Research, The University of Hong Kong, Hong Kong, China; u3008234@connect.hku.hk (X.-Y.L.); ymtsui@hku.hk (Y.-M.T.); tamivan@pathology.hku.hk (I.K.-K.T.); lipm@pathology.hku.hk (P.-M.L.); garycch@pathology.hku.hk (G.C.-H.C.); joyce@pathology.hku.hk (J.M.-F.L.); iolng@hku.hk (I.O.-L.N.); 2Department of Pathology, School of Clinical Medicine, The University of Hong Kong, Hong Kong, China

**Keywords:** mitochondrial copy number, target-panel sequencing, liver cancer, HCC

## Abstract

**Simple Summary:**

Mitochondria are important organelles in the human body, and they act as a source of chemical energy for different cellular processes. Multiple copies of mitochondria are normally detected in cells, and their copy number has also been reported to be aberrated in cancers, including liver cancer. Target-panel sequencing is frequently used in oncology studies, but the optimal pooling ratio of nuclear and mitochondrial probes has remained elusive. Therefore, it is necessary to determine the physiological mitochondrial DNA copy number in non-liver cancer subjects, which can serve as a useful reference for deciding the desirable ratio of nuclear and mitochondrial probes for a less biased target enrichment of nuclear and mitochondrial genomes and hence their even representation. In addition, it can also be used as a physiological basis that can distinguish normal physiological variations from the pathological conditions in liver cancer.

**Abstract:**

Mitochondria generate energy to support cells. They are important organelles that engage in key biological pathways. The dysfunction of mitochondria can be linked to hepatocarcinogenesis, which has been actively explored in recent years. To investigate the mitochondrial dysfunction caused by genetic variations, target-panel sequencing is a flexible and promising strategy. However, the copy number of mitochondria generally exceeds nuclear DNA, which raises a concern that uneven target enrichment of mitochondrial DNA (mtDNA) and nuclear DNA (ncDNA) in target-panel sequencing would lead to an undesirably biased representation of them. To resolve this issue, we evaluated the optimal pooling of mtDNA probes and ncDNA probes by a series of dilutions of mtDNA probes in both genomic DNA (gDNA) and cell-free DNA (cfDNA) samples. The evaluation was based on read count, average sequencing depth and coverage of targeted regions. We determined that an mtDNA:ncDNA probe ratio of around 1:10 would offer a good balance of sequencing performance and cost effectiveness. Moreover, we estimated the median physiological mtDNA:ncDNA copy ratio as 38.1 and 2.9 in cfDNA and gDNA samples of non-liver cancer subjects, respectively, whereas they were 20.0 and 2.1 in the liver cancer patients. Taken together, this study revealed the appropriate pooling strategy of mtDNA probes and ncDNA probes in target-panel sequencing and suggested the normal range of physiological variation of the mtDNA:ncDNA copy ratio in non-liver cancer individuals. This can serve as a useful reference for future target-panel sequencing investigations of the mitochondrial genome in liver cancer.

## 1. Introduction

Mitochondria are multifunctional organelles, and their functions are essential for life [1,2]. They act as powerhouses to synthesize ATP and participate in other processes involving inflammation, metabolism, cell death, cellular transformation, and cell fate determination [3,4,5,6]. The deficiency of mitochondrial functions caused by its related gene mutations and copy number variations can lead to dysfunction of metabolic pathways, unbalance of redox, avoidance of apoptosis, and therapy resistance, contributing to multiple types of tumor initiation [7,8], including hepatocarcinogenesis [9,10,11,12].

The D-loop region of mitochondria is the hotspot of somatic mutations and the most susceptible site to oxidative damage [13,14]. A reduction in the copy number of mitochondrial DNA (mtDNA) was correlated with hepatocellular carcinoma (HCC) progression [15,16], and its non-invasive application in blood-based samples could be a potential biomarker for HCC diagnosis and prognosis [16,17]. Exploration of mitochondrial genetic variations, e.g., mutations or structural alterations in the mitochondrial genome [13,18,19], would promote an understanding of the roles of mitochondria in HCC development.

Liquid biopsy using circulating cell-free DNA (cfDNA) is a promising and non-invasive strategy for detecting genomic alternations [14]. Target-panel sequencing (target-seq) focuses on specific genomic areas of interest instead of the whole genome, providing a flexible yet focused strategy for in-depth molecular profiling at a high read coverage. To investigate the status of the mitochondrial genome using target-seq in liquid biopsy, somatic mutations are commonly detected by subtracting mutations detected in peripheral blood mononuclear cells (PBMCs) from the corresponding plasma cfDNA. However, mtDNA copy number may vary according to the cell type composition in human blood [20,21,22,23]. More importantly, the mtDNA and nuclear DNA (ncDNA) do not typically exist in an equal copy ratio, and the difference can be as large as >100-fold [21,24]. Given the pivotal roles of the mtDNA genome in HCC [16,25], target-seq of HCC patients commonly involves the combination of mtDNA and ncDNA components in the panel design. In view of the substantially unequal abundance of mtDNA and ncDNA, this may likely result in a significant disparity in the availability of their DNA content, leading to concerns about their relative performance in target enrichment and the undesirable bias in their uneven genomic representation. It is of particular concern that an extremely high coverage of the mtDNA panel may in turn deteriorate the coverage of the ncDNA counterpart. Thus, it is critical to investigate the optimal pooling ratio of mtDNA and ncDNA probes for target enrichment in target-seq. Moreover, pathological conditions in HCC may impose an abnormal copy number aberration in mtDNA and modify the difference in mtDNA and ncDNA content. Therefore, there is an urgent need to evaluate the physiological disparity between mtDNA and ncDNA, which can provide the physiological basis that could be applicable to distinguish normal physiological variations from the pathological conditions.

In this study, we aimed to estimate the bias in target enrichment of mtDNA and ncDNA in both plasma cfDNA and genomic DNA, and their influence on the performance of target-seq. Furthermore, our findings in non-HCC subjects will provide a useful physiological reference for the non-pathological range of variation in the copy number ratio between mtDNA and ncDNA.

## 2. Materials and Method

### 2.1. Sample Collection, DNA Extraction, Library Construction

Twenty non-HCC individuals and ten HCC patients aged >18 years were recruited from Li Ka Shing Faculty of Medicine, which was approved by the Institutional Review Board of the University of Hong Kong (UW 19-667). Plasma and peripheral blood mononuclear cells (PBMCs) were extracted from whole blood collected in EDTA tubes. Referring to our previous study [26], genomic DNA (gDNA) was extracted from PBMC samples. For plasma, following our established method (Nivo study), cfDNA was extracted from 1.5–2 mL plasma by the Cobas cfDNA Preparation Kit (Roche, Basel, Switzerland, 07247737190) according to the manufacturer’s instructions. The quantitation of eluted cfDNA was estimated by Qubit dsDNA HS Assay Kit (Thermo Fisher Scientific, Waltham, MA, USA), and the DNA fragment size was analyzed by the High-Sensitivity NGS Fragment Analysis Kit (Agilent, Santa Clara, CA, USA, DNF474).

### 2.2. Next-Generation Sequencing and Data Preprocessing

For the target-seq experiment, gDNA from PBMC and cfDNA from plasma were used to prepare for DNA libraries. All the libraries were prepared based on the protocol of the KAPA Hyper Prep Kit Technical Sheet (KR0961). DNA was fragmented by Diagenode Bioruptor Pico system, and the fragmented DNA was performed end-repair, A-tailing at the 3′ end, adaptors (with indexes) ligation at the terminal ends. The adapter ligated library with a size range of 350 bp–700 bp was selected by dual-SPRI method. PCR was performed, and the products were validated by Agilent Fragment Analyzer.

We utilized the KAPA HyperExplore nuclear panel (947 genes) and KAPA HyperChoice mito panel for target enrichment of nuclear and mitochondrial genes, respectively. Notably, due to the unequal copy number of ncDNA and mtDNA in the human genome, equal pooling of the 2 panels may likely result in an unequal representation of the nuclear and mitochondrial genes, leading to significant bias favoring the generation of mitochondrial reads. To evaluate the optimal pooling ratio of the mito and nuclear panels, we generated target enrichment probe mixtures with different mtDNA:ncDNA probe ratios by volume (1:5, 1:10, 1:20, 1:50 and 1:100), as suggested by the manufacturer. For instance, an mtDNA:ncDNA probe ratio of 1:5 refers to the mixing of 1 unit of mtDNA probes and 5 units of ncDNA probes. Target enrichment was conducted according to the manufacturer’s protocols and workflow (KAPA HyperCap Workflow v3.2). Before hybridization, libraries were normalized and combined with different indices into a single pool of 1.5 micrograms in total prior to enrichment. The pooled DNA libraries were mixed with capture probes of targeted regions. The enriched libraries on the beads were then amplified by PCR. The enriched libraries were validated by Agilent Bioanalyzer, Qubit and qPCR for quality control analysis. Illumina NovaSeq 6000 was applied for paired-end 151 bp sequencing. The preparation of the DNA library and sequencing were conducted by the Centre for PanorOmic Sciences of the University of Hong Kong. Target-seq data were subjected to quality control and adapter trimming by FASTQC and Trimmomatic.

### 2.3. Data Processing and Sequencing Quality Estimation

Target-seq data were subjected to quality control and adapter trimming by FASTQC and Trimmomatic. Paired-end reads were aligned to human reference genome (hg38) by BWA. Also, PCR duplicates were removed by PICARD. Read count, sequencing coverage, and depth were estimated by bamdst. The mtDNA:ncDNA copy ratio was estimated by the average sequencing depth of the mito panel divided by that of the nuclear panel and multiplied by the corresponding dilution factor.

## 3. Results

### 3.1. Sequencing Read Count under Different Dilutions of mtDNA Probes

To determine the optimal pooling ratio of mtDNA probes and ncDNA probes for target-seq and to estimate the physiological copy number ratio between mtDNA and ncDNA, we recruited 10 non-HCC individuals and collected their paired PBMC and plasma samples to isolate genomic DNA (gDNA) and cell-free DNA (cfDNA), respectively. We generated a series of target enrichment probe mixtures with different mtDNA:ncDNA probe ratios (1:5, 1:10, 1:20, 1:50 and 1:100) (Figure 1A).

After mapping the sequencing reads to the human reference genome and removing PCR duplicates, we calculated the read count of the mtDNA and ncDNA panels separately in both gDNA (PBMC samples) and cfDNA (plasma samples). Generally, the read count of the nuclear panel for gDNA was higher than that of cfDNA due to the relatively lower availability of DNA in plasma (Figure 1B). In contrast, the read count of the mito panel was just similar between gDNA and cfDNA, indicating that the small-size mito panel performed almost equally at different availabilities of DNA (Figure 1B). In fact, the size of the mito panel is ~16.5 kb, whereas the nuclear panel is close to 2.6 Mb. The approximately 164-fold difference between the nuclear and mito panels likely exacerbated the demand for input DNA availability and led to the apparent inequality of the achieved read count in gDNA and cfDNA. Similar trends were found in individuals, according to which the read counts of the mito panel were generally low at different mtDNA:ncDNA probe pooling ratios, but there was a reduction trend in the read count towards a high dilution ratio of 1:100 in both cfDNA (lower extent of reduction) and gDNA (higher extent of reduction) (Figure 1C). On the other hand, the read counts of the nuclear panel were relatively stable at different mtDNA:ncDNA probe pooling ratios, and the difference between cfDNA and gDNA samples was substantial (Figure 1D). Taken together, the performance of the mito panel was more sensitive to the choice of the mtDNA:ncDNA probe pooling ratio, with a higher dilution ratio eliciting a greater impact on the deterioration of the sequencing performance.

### 3.2. Average Sequencing Depth under Different Dilutions of mtDNA Probes

The sequencing depth (total sequencing length of reads divided by panel size) covering the regions of interest could greatly influence the efficiency of identification of sequence variants, and adequate coverage facilitates the sensitivity and precision of genetic variation detection. To find the optimal pooling ratio between mtDNA probes and ncDNA probes, different pooling ratios were generated, and the average sequencing depths at the target panels were calculated as an indicator of sequencing performance. The average sequencing depth of the nuclear panel was invariable in both gDNA and cfDNA at different pooling ratios (Figure 2A). However, a higher level of average sequencing depth was detected in gDNA (median: 1149X) compared to that in cfDNA (median: 125X). This may likely reflect the fact that the lower availability of DNA in cfDNA could limit the achievable sequencing depth. For the mito panel, the average sequencing depth progressively decreased towards a higher ratio, indicating that the choice of dilution ratio can make an impact on mtDNA sequencing. Even though variations were observed, the coefficient of variation in the average sequencing depth was relatively stable in gDNA and cfDNA samples at different dilutions (Figure 2B). In general, the performance of gDNA was slightly higher than that of cfDNA, but beyond a ratio of 1:20, their outcomes were almost the same. There was an intrinsic difference in panel size between mito and nuclear panels. To evaluate the optimal pooling strategy for our analysis, we calculated the fold change of the average sequencing depth between nuclear and mito panels at different mtDNA:ncDNA probe pooling ratios (Figure 2C). Compared to gDNA samples, cfDNA samples were more tolerant to different pooling ratios, and both nuclear and mito panels achieved relatively more similar outcomes. In order to aim for a similar performance for the nuclear and mito panels, we should ideally target the mtDNA:ncDNA probe dilution ratio to be no more than 1:10 (choosing a ratio of 1:10 can minimize the cost of purchasing mtDNA probes, without a marked deterioration of the mtDNA sequencing performance).

### 3.3. Coverage of Targeted Regions under Different Dilutions of mtDNA Probes

In addition to the sequencing depth, the % coverage of target panel (regions of interest) at the designated minimum sequencing depth was estimated. We applied the coverage of targeted regions as another indicator for the estimation of the appropriate dilution of mtDNA probes. Under a lower sequencing depth requirement (≥10× and ≥30×), coverage of both nuclear and mito panels was satisfactory (>98%) for both gDNA and cfDNA samples (Figure 3A,B). The situation for the mito panel was similar, except the performance markedly deteriorated at a high pooling ratio (especially 1:100). Under a higher sequencing depth (≥100× and ≥150×), the coverage of the nuclear panel was invariable for both gDNA and cfDNA samples (Figure 3C,D). The performance of the mito panel for both gDNA and cfDNA samples deteriorated towards a higher pooling ratio, but gDNA achieved better coverage in general.

### 3.4. Normal Physiological Range of Variation in mtDNA:ncDNA Copy Ratio

By taking into account the physiological variation of the average sequencing depth (mito and nuclear panels) in the non-HCC sample cohort and the observed variations under different dilutions of mtDNA probes, we reconstructed the normal physiological range of variation in the mtDNA:ncDNA copy ratio. The median estimated ratios of mtDNA and ncDNA contents in cfDNA and gDNA samples were 34.4 (range: 10.8–173.3) and 5.8 (range: 2.6–9.2), respectively (Figure 4A). We further sequenced 10 additional subjects using an mtDNA:ncDNA probe ratio of 1:10 for validation. Similarly, we reconstructed the normal physiological mtDNA:ncDNA copy ratio, with median values for cfDNA and gDNA determined to be 24.4 (range: 10.4–70.3) and 2.4 (range: 2.0–3.0), respectively (Figure 4B). The two sets of non-HCC subjects had consistent normal physiological mtDNA:ncDNA copy ratios, and, importantly, they collectively suggested a reference range of normal physiological variation (Figure 4C). In addition, we believe gDNA demonstrated a narrower range of variation, whereas cfDNA exhibited a greater extent of disparity between subjects (Figure 4A–C). This may imply that a greater deviation from this estimated normal physiological range would be required for cfDNA samples in order to indicate genuine pathological conditions, as compared to the gDNA counterparts.

### 3.5. Physiological Range of Variation in mtDNA:ncDNA Copy Ratio in HCC Patients

Given that we estimated the normal physiological mtDNA:ncDNA copy ratio using non-HCC subjects and the reasonable choice of the mtDNA:ncDNA probe dilution ratio to be 1:10, we further investigated the physiological range of the mtDNA:ncDNA copy ratio by sequencing another clinical cohort of 10 HCC patients. Notably, a significantly lower mtDNA:ncDNA copy ratio was found in HCC patients compared to non-HCC subjects in both cfDNA and gDNA (Figure 5A,B). Our findings align with previous studies showing that a reduction of the mtDNA copy number was detected in HCC compared to normal [15,16].

## 4. Discussion

Mitochondria play an important role in metabolically supporting tumor development, and many studies [27,28] focus on investigating the functional consequence of mitochondrial genetic variation. However, there are endogenous copy number differences between mtDNA and ncDNA, and the copy number of mtDNA also varies in cell types. For DNA library preparation in targeted sequencing including both nuclear and mito gene panels, the optimal pooling ratio of mtDNA probes and ncDNA probes has rarely been explored. In our study, we performed different dilutions of mtDNA probes (relative to ncDNA probes) for target enrichment on gDNA and cfDNA samples to find the optimal ratio that is likely to be valid under usual circumstances and could be applicable to a general usage. Moreover, our study also provided a normal range of mtDNA:ncDNA ratios in non-HCC subjects, which could be treated as a physiological reference of their normal interval of variation.

The sequencing data output of ncDNA was not severely affected by the choice of dilution of mtDNA probes in both gDNA and cfDNA samples, whereas that of mtDNA was substantially and adversely influenced at a high dilution ratio. When pinpointing the performance of the mito panel, at a higher dilution of mtDNA probes, the quality of sequencing data, including the read count, average sequencing depth, and coverage of the target panel, was consistently deteriorated. On the other hand, at a lower dilution of mtDNA probes, the quality of sequencing data was generally satisfactory. Taken together, we recommend the optimal pooling ratio of mtDNA:ncDNA probes to be at around 1:10. As pathological changes of mtDNA are common in human cancers [15,16,29,30], it is therefore advisable to systematically estimate the normal physiological range of variation in the mtDNA copy number relative to ncDNA. Based on our empirical estimation, compared to gDNA samples, cfDNA samples would require a higher extent of deviation of the mtDNA:ncDNA copy ratio from the estimated normal range in order to differentiate pathological conditions at a high confidence.

The physiological mtDNA copy number varies widely in individuals. It is associated with relevant cellular components in blood, especially circulating immune cells [21], which depend on the status of individuals. Furthermore, many studies have reported that the mtDNA copy number is also affected by age, gender, general health, and other factors [31,32,33]. Apart from individual factors, pre-analytical factors can also influence the measurement of the mtDNA copy number. The time between blood withdrawal and cell separation, DNA extraction methodology, kit used for DNA isolation, and testing platform could possibly introduce variations in mtDNA quantification [21,34,35]. Concerning the above factors, previous studies have reported a great variety of mtDNA copy numbers, including high-level ones (mean value: >200) [22], middle-level ones (mean value: 60–100) [36,37], and low-level ones (mean value: <10) [38,39,40].

Our study utilized a target-seq strategy to investigate the physiological variation in the mtDNA:ncDNA copy ratio in gDNA and cfDNA samples from liver-cancer and non-liver-cancer subjects. However, there are several limitations to our study that awaits further validation by future studies. First, although our findings achieved statistical significance, confirmation using a larger sample cohort and perhaps in other non-Chinese ethnicities will be worth pursuing. Second, we carried out our investigations using target-seq because it is a popular and cost-effective sequencing modality that has been utilized in many oncology studies. Nevertheless, the choice of target-seq platform, targets in the gene panel, as well as the target enrichment process may introduce variability in capture efficiency that could possibly lead to bias in estimation, which should ideally be conducted by methods without a target enrichment process, e.g., whole-genome sequencing. Moreover, gDNA from PBMC samples was tested in our comparison. Given that PBMC may vary in the composition of immune cells, phenotype and activation status in different cancer and non-cancer subjects, gDNA from other sources should also be tested for a better estimation.

## 5. Conclusions

Our study has explored the optimal strategy for pooling mtDNA probes and ncDNA probes in a target-seq experiment and provides useful guideline for identifying pathological conditions that are potentially applicable to HCC, based on an mtDNA copy number perspective. In general, a significantly lower mtDNA:ncDNA copy ratio was found in HCC patients compared to non-HCC subjects in both cfDNA and gDNA. Our findings can serve as a useful reference for future target-panel sequencing investigations of the mitochondrial genome in liver cancer.

## Figures and Tables

**Figure 1 cancers-16-03012-f001:**
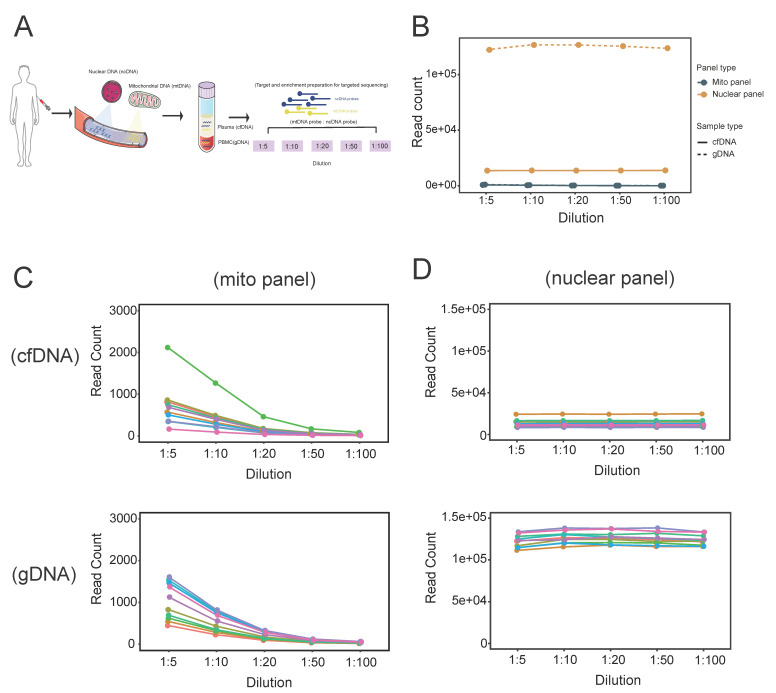
Evaluation of sequencing read count for optimal pooling of mtDNA probes and ncDNA probes. (**A**) Strategy of pooling mtDNA probes and ncDNA probes for target-seq in plasma (cfDNA) and PBMC (gDNA) samples. (**B**) Overall read count of the mito panel and nuclear panel in cfDNA and gDNA at different pooling ratios. Read count of mito panel (**C**) and nuclear panel (**D**) for cfDNA and gDNA samples at individual level.

**Figure 2 cancers-16-03012-f002:**
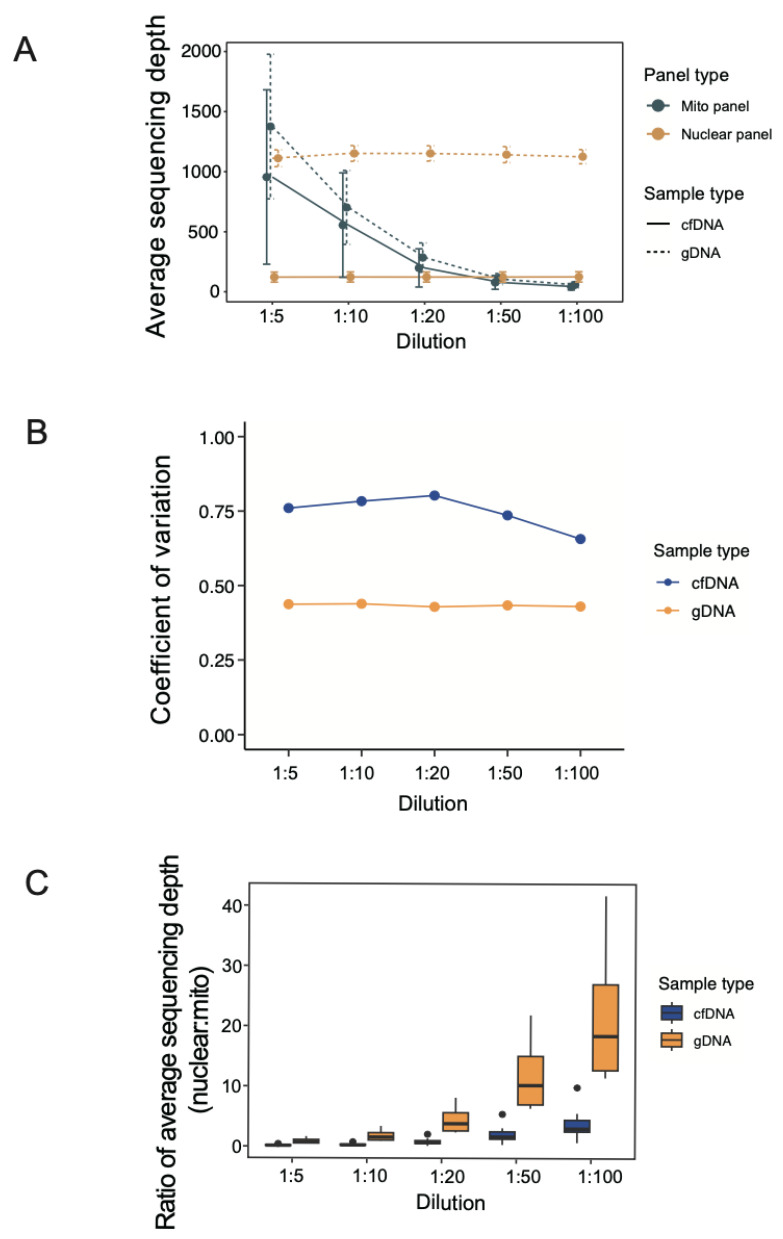
Evaluation of average sequencing depth for optimal pooling of mtDNA probes and ncDNA probes. (**A**) Average sequencing depth of mito panel and nuclear panel in cfDNA and gDNA. The height of the error bar indicates the standard deviation of depth. (**B**) Coefficient of variation of average sequencing depth in cfDNA and gDNA. (**C**) Fold change of average sequencing depth between nuclear and mito panels under different pooling ratios.

**Figure 3 cancers-16-03012-f003:**
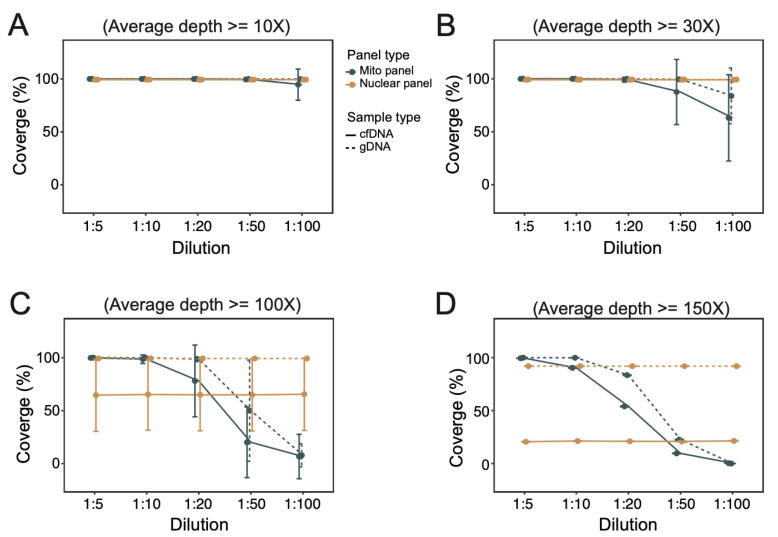
Evaluation of coverage of target regions for optimal pooling of mtDNA probes and ncDNA probes. Coverage of targeted regions of mito panel and nuclear panel in cfDNA and gDNA at different minimum sequencing depth thresholds, namely ≥10× (**A**), ≥30× (**B**), ≥100× (**C**), and ≥150× (**D**).

**Figure 4 cancers-16-03012-f004:**
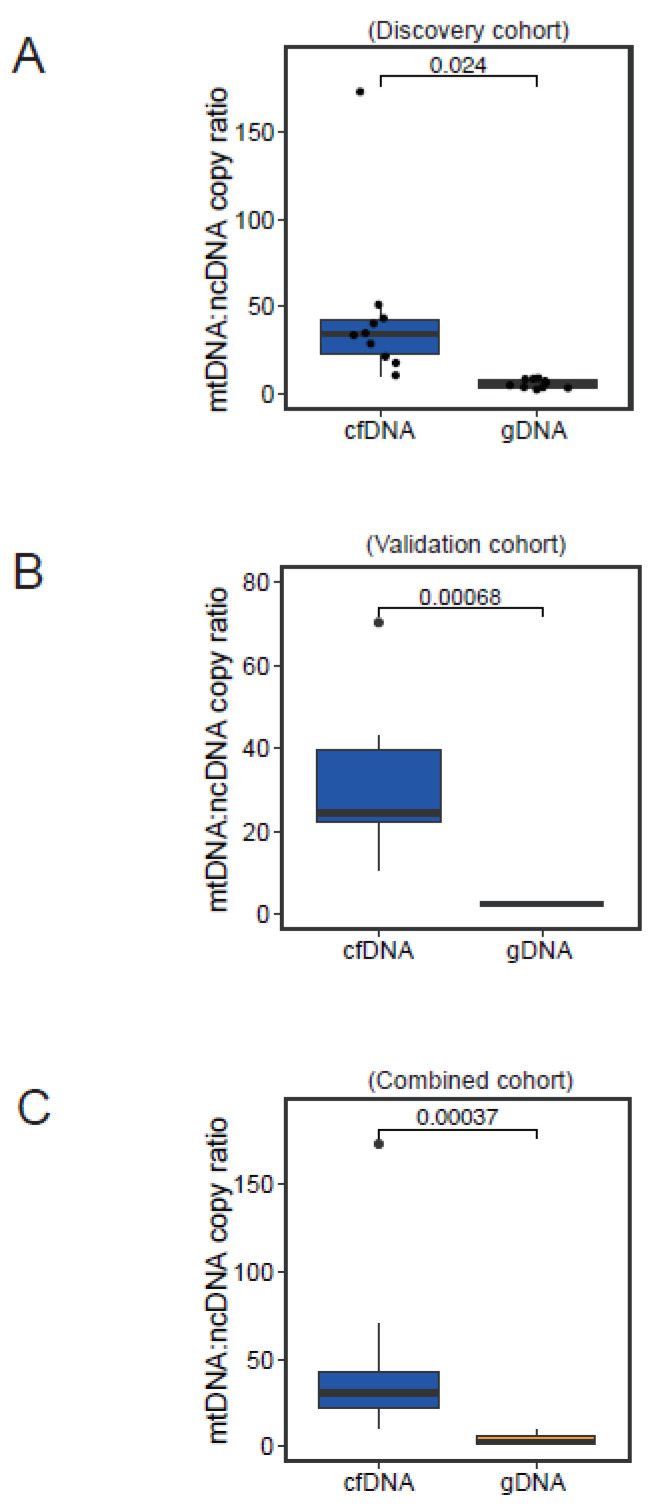
Estimated copy ratio between mtDNA and ncDNA in cfDNA and gDNA. The boxplot summarizes the values of the mtDNA:ncDNA copy ratio in the discovery cohort (N = 10) (**A**), validation cohort (N = 10) (**B**), and combined cohort (N = 20) (**C**).

**Figure 5 cancers-16-03012-f005:**
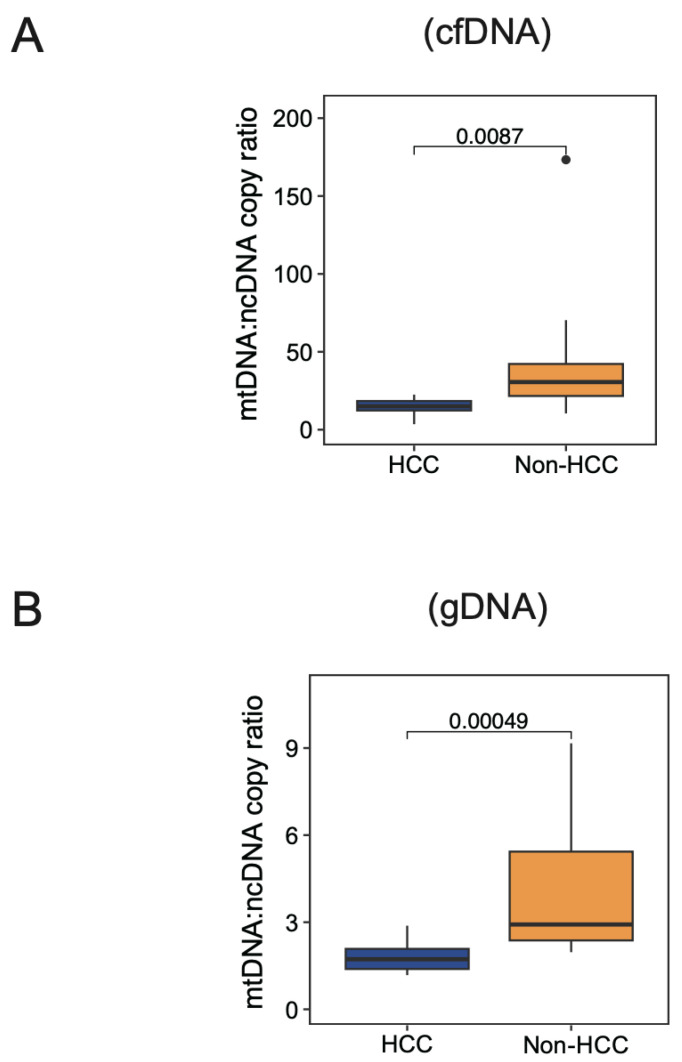
Investigation of mtDNA:ncDNA copy ratio in liver cancer subjects. Comparison of reconstructed physiological mtDNA:ncDNA copy ratio in cfDNA (**A**) and gDNA (**B**) of HCC patients (N = 10) and healthy subjects (N = 20).

## Data Availability

The data presented in this study are available upon request to the corresponding author.

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
