# Peer review of "Resolution of Optimal Mitochondrial and Nuclear DNA Enrichment in Target-Panel Sequencing and Physiological Mitochondrial DNA Copy Number Estimation in Liver Cancer and Non-Liver Cancer Subjects"

_cancers, 2024, doi:10.3390/cancers16173012_

Round 1

Reviewer 1 Report

Comments and Suggestions for Authors

The work of Xueying Lyu and collaborators is interesting, well developed and presented

Some suggestions:

Was the protocol approved by an ethics or research committee? What is its registry?

I suggest expanding the discussion, what is its conclusion? What are the perspectives of the study?

What were the limitations of the study?

Reviewer 2 Report

Comments and Suggestions for Authors

The manuscript of “Resolution of optimal mitochondrial and nuclear DNA enrichment in target-panel sequencing and physiological mitochondrial DNA copy number estimation in liver cancer and nonliver cancer healthy subjects” by Xueying Lyu and co-authors aims to develop the optimal strategy of pooling mtDNA and ncDNA probes in target-panel sequencing and to provide useful guidelines that could be applicable to distinguish variations in plasma circulating cell-free DNA (cfDNA) and genomic (PBMC) DNA samples in healthy subjects from the abnormal range of mtDNA:ncDNA ratio in patients with hepatocellular carcinoma (HCC). In the study, twenty healthy individuals and ten HCC patients aged >18 years were included. Illumina NovaSeq 6000 was applied for paired-end 151 bp sequencing. The authors found that 1) the optimal pooling ratio of mtDNA:ncDNA probes was approximately 1:10, and 2) the medians physiological mtDNA:ncDNA copy ratio were 38.1 and 2.9 in cfDNA and gDNA samples of healthy subjects, respectively, whereas they were 20.0 and 2.1 in the HCC patients.

The relevance of the research topic is due to the sharp increase in the number of liver cancer throughout the world and the urgent need to find new methods for its early diagnosis and effective treatments. The manuscript is, in general, well written and contributes to the clinical validation of target-panel sequencing of mitochondrial genome for its application to disparate the most common type of liver cancer, hepatocellular carcinoma, and some physiological variations. The experimental design was well thought out. However, there are some comments that deserve further attention of the authors.

Comments:

1. The Introduction section could be improved. In particular, a number of the authors’ statements are unsubstantiated and require appropriate references. For example, Lines 44-45: “Liquid biopsy using circulating cell-free DNA (cfDNA) is a promising and non-invasive strategy for detecting genomic alternations.”; Lines 54-55: “target-seq of HCC patients commonly involves the combination of mtDNA and ncDNA components in the panel design.”, and many others.

2. There are some aberrations in the Abstract and Introduction section that are not deciphered at the first mention. This makes it difficult to understand the text. For example, Line 25: HCC patients; Line 40: HCC, etc.

3. The authors wrote that “Exploration of mitochondrial genetic variations would promote the understanding of the roles of mitochondria in HCC development”. Indeed, targeted gene sequencing panels can be useful tools for analyzing specific mutations in a given sample. However, this information was not provided. Please, clarify.

4. It is known that quantitative polymerase chain reaction (qPCR) analysis is widely used to study changes in the mtDNA:ncDNA copy ratio in plasma cfDNA and peripheral blood mononuclear cell gene DNA. Please clarify what advantages the target-seq experiments may provide.

4. The authors compared the ranges of variation in mtDNA:ncDNA copy ratio in plasma cfDNA and PBMC gDNA in healthy subjects and HCC patients. However, PBMCs have a different composition, phenotype and activation status in healthy subjects and HCC patients. As is known, PBMCs are important immune system components, consisting of T cells, B cells, NK cells, and monocytes, and they can reflect systemic immune response during cancer progression. The copy number of mtDNA also varies in different types of PBMCs. Authors should provide some clinical blood test results in healthy subjects and HCC patients, and/or discuss these issues in more detail in the Discussion.

5. The Discussion section is poorly written and could be improved.

Comments on the Quality of English Language

Minor editing of the English text is required.

Reviewer 3 Report

Comments and Suggestions for Authors

This is a very specialised paper studying the optimal enrichment strategy for target enrichment based sequencing. I have the following comments:

1. As far as I understand the provided information is relevant for the use of KAPA HyperExplore nuclear panel (947 genes) and KAPA HyperChoice 92
mito panel based enrichment. That should be more clearly indicated, since the use of different nuclear enrichment panels would potentially lead to different results.

2. The average sequencing depth shown in Fig. 2 has for mtDNA even for gDNA a very big SD value. How this can be explained?

3. Why in Fig. 3D (average depth > 150x) the SD values are much lower than in Fig. 3C (average depth > 100x)?

4. The physiological mtDNA copy number in gDNA from blood samples is according to literature data around 100. It is unclear how the authors obtained 5.8 (Line 201), since this value obviously depends on the degree of target enrichment. For obtaining the real physiological value from NGS data no target enrichment (WGS condition) should be made.

5. To me it is not clear how this very specialized methodical paper relates to liver cancer. How the authors explain their findings of differences in mtDNA copy number in gDNA from whole blood between liver cancer patients and controls?

Comments on the Quality of English Language

None.

Round 2

Reviewer 2 Report

Comments and Suggestions for Authors

The manuscript has been revised. English language is fine; no issues were edtected.

Reviewer 3 Report

Comments and Suggestions for Authors

The authors have addressed all of my concerns.